# Validation of Embedded State Estimator Modules for Decentralized Monitoring of Power Distribution Systems Using IoT Components

**DOI:** 10.3390/s22062104

**Published:** 2022-03-09

**Authors:** Rosvando Marques Gonzaga Junior, Sergio Márquez-Sánchez, Jorge Herrera Santos, Rodrigo Maximiano Antunes de Almeida, João Bosco Augusto London Junior, Juan Manuel Corchado Rodríguez

**Affiliations:** 1São Carlos School of Engineering, University of São Paulo, São Carlos 13566-590, Brazil; jbalj@sc.usp.br; 2BISITE Research Group, Computer and Automation Department, University of Salamanca, Calle Espejo s/n. Edificio Multiusos I+D+i, 37007 Salamanca, Spain; smarquez@usal.es (S.M.-S.); jorgehsmp@usal.es (J.H.S.); corchado@usal.es (J.M.C.R.); 3Air Institute, IoT Digital Innovation Hub (Spain), 37188 Salamanca, Spain; 4Institute of Systems Engineering and Information Technologies, Federal University of Itajubá, Itajubá 37500-903, Brazil; rodrigomax@unifei.edu.br

**Keywords:** distribution systems, smart meters, advanced measurement infrastructure, state estimation, embedded systems, Internet of Things, edge computing

## Abstract

Recent theoretical studies demonstrate the advantages of using decentralized architectures over traditional centralized architectures for real-time Power Distribution Systems (PDSs) operation. These advantages include the reduction of the amount of data to be transmitted and processed when performing state estimation in PDSs. The main contribution of this paper is to provide lab validation of the advantages and feasibility of decentralized monitoring of PDSs. Therefore, this paper presents an advanced trial emulating realistic conditions and hardware setup. More specifically, the paper proposes: (i) The laboratory development and implementation of an Advanced Measurement Infrastructure (AMI) prototype to enable the simulation of a smart grid. To emulate the information traffic between smart meters and distribution operation centers, communication modules, that enable the use of wireless networks for sending messages in real-time, are used, bridging concepts from both IoT and Edge Computing. (ii) The laboratory development and implementation of a decentralized architecture based on Embedded State Estimator Modules (ESEMs) are carried out. ESEMs manage information from smart meters at lower voltage networks, performing real-time state estimation in PDSs. Simulations performed on a real PDS with 208 buses (considering both medium and low voltage buses) have met the aims of this paper. The results show that by using ESEMs in a decentralized architecture, both the data transit through the communication network, as well as the computational requirements involved in monitoring PDSs in real-time, are reduced considerably without any loss of accuracy.

## 1. Introduction

Smart Grid (SG) initiatives have increased the complexity of Power Distribution Systems’ (PDSs) operation and control. This complexity emerges from additional features, such as subnetworks that can operate autonomously, the penetration of new renewable energy sources in the Low Voltage (LV) and Medium Voltage (MV) distribution networks, energy storage technologies, the connection of electric vehicles to the PDSs, etc.

These SG initiatives have created new requirements for the operation of distribution networks. The needs emerging from the connection of electric vehicles to PDSs have been the focus of recent papers [1,2,3,4].

These needs may result in higher power demand, an increase in short-circuit currents, exceeded voltage limits, and a reduced lifespan of the power equipment (e.g., transformers). Therefore, advanced control actions must be carried out by Distribution System Operators (DSOs) [5,6].

An accurate estimate of the real-time operating condition of PDSs is essential for performing advanced control actions in such systems. However, historically, the limited amount of real-time measurements in PDSs have prevented the deployment of State Estimators (SEs) in these networks [7,8,9]. Consequently, the real-time operating condition of PDSs has been usually determined through the statistical characterization of their loads performed by a process called load aggregation [10,11,12]. Nevertheless, in recent years, the development of new equipment for real-time measurements, such as micro-PMUs (Phasor Measurement Units) and Smart Meters (SMs), has motivated the development of Distributions System State Estimators (DSSEs) [13].

Although this equipment has contributed to the development of DSSEs, the increase in the number of real-time measurements has made a further investigation into DSSE development necessary because of the increase in the volume of data to be transmitted to the Distribution System Operation Centers (DOCs) [14].

Another challenge to the deployment of DSSEs, which is now becoming greater, is the large scale of PDSs. Note that the MV distribution networks already involve large-scale networks, usually composed of tens of thousands of three-phase unbalanced nodes. The addition of LV distribution networks to where the SMs are being installed within the concept of Advanced Measurement Infrastructure (AMI), will raise this scale even more, to hundreds of thousands of nodes.

In this context, where the volume of data to be transmitted to the DOCs increases and the scale of PDSs networks expands, decentralized architectures for PDS monitoring appear to be a promising proposal. Unlike in decentralized architectures, in conventional centralized architectures, all the information must be sent to DOCs and processed. Thus, the processing power and bandwidth of the communication network must be very high in a centralized architecture if real-time performance is to be tractable [7].

Among the DSSEs developed for decentralized architectures, multiarea DSSEs are oriented towards achieving computational performance and scalability. The decentralized multilevel approach presented in [15] is also an interesting proposition. It integrates Cloud Computing and the Internet of Things (IoT) to deal with the different elements of the PDSs (MV and LV networks) simultaneously. As the approach uses a Cloud Computing framework, it is attractive for integrating aggregators and decentralized local markets. However, it requires a physical structure of data concentrators to be installed in all MV/LV transformers, and cloud service latency was not considered, which may hamper real-time integration with DOCs.

Advances in communication networks aimed at implementing the IoT ecosystem (such as Sigfox, Narrow Band IoT, Long Range, among others) combined with a greater ability to process embedded electronic devices, enable the development of electronic modules that can operate independently using the concept of edge computing. Based on these advances, in [7], decentralized architectures for PDS monitoring were proposed and compared to the centralized architecture that is traditionally used in control centers. Although not conclusive, the theoretical results demonstrated the advantages of decentralized architectures, from the point of view of reducing the amount of data to be transmitted and processed when performing state estimation.

The main contribution of this paper is to provide conclusive evidence for the advantageousness and feasibility of the decentralized monitoring of PDSs, since only theoretical evidence was published in [7]. The focus of this paper is laboratory validation; an advanced trial is presented, that emulates realistic conditions.

To provide conclusive evidence for the advantageousness and feasibility of the decentralized monitoring of PDSs, this paper proposes:1.The laboratory development and implementation of an AMI, making use of embedded modules such as NVIDIA Jetson Nano and Raspberry PI 3, coupled with wireless communication modules for the development of wireless networks, over which messages may be sent and received in real-time, using the concepts of IoT and Edge Computing. The function of each module in the proposed laboratory infrastructure is presented in Section 4. Observe that the objective is not to compare the performance of the electronic components, but to implement an AMI in a laboratory, enabling the simulation of a smart grid with regard to information traffic between the SMs, represented by Raspberry PI 3, and DOCs, represented by the Things Board Platform;2.The laboratory development and implementation of a decentralized architecture based on Embedded State Estimator Modules (ESEMs) that manage information from SMs in LV networks, performing real-time state estimation in PDSs. Each ESEM is equipped with a high-performance processor with a Linux operating system to enable matrix analysis related to the Weighted Least Squares (WLS) state estimator.

The remaining part of this paper is organized as follows: Section 2 presents an overview of the related literature. Section 3 presents the communications of the networks and protocols related to IoT technologies. Section 4 details the system design and the developed devices. Section 5 presents the data analysis as well as the union of the different models for the final solution. Finally, the last section presents some conclusive remarks and future lines of research.

## 2. Real-Time Monitoring of Power Distribution Systems

### 2.1. Power Distribution Systems

The purpose of PDSs is to meet the customer´s energy demand, after receiving the bulk energy from transmission (or sub-transmission) lines (KERSTING, 2011), and comprises: (i) distribution substations, which are fed by one or more transmission (or sub-transmission) lines; (ii) primary distribution lines or feeders (known as the MV networks); (iii) distribution transformers, which connect the MV networks to the LV networks; (iv) secondary distribution circuits (known as the LV networks), and (v) customers’ connections.

The historic passive behavior of PDSs has changed in the last years mainly because of the development of smart grid features, such as massive installation of intermittent generation/load, use of storage devices, and diffusion of electric vehicles. Therefore, PDSs have become highly complex systems which makes it necessary to efficiently monitor their operating conditions.

The SE is the fundamental tool for the efficient real-time monitoring of electric networks, which has the function of estimating, in real-time, the network state variables (usually the nodal complex voltages) by processing a set of redundant noisy measurements. Through the analysis of these state variables, the operating condition of an electric network can be determined.

### 2.2. State Estimation

Despite the consolidated position of the WLS SE for transmission systems, specialized algorithms have been developed to perform state estimation in PDSs [9]. However, due to the particularities of PDS networks, specialized DSSEs often rely on the simplifications and approximations of the measurement model which make it difficult to generalize their results [16]. Therefore, some researchers have proposed new versions of the WLS estimator for PDSs, such as the practical WLS multi-phase DSSE proposed in [17]. It is a sparse and numerically stable implementation of the WLS SE, which does not require any additional assumptions from the traditional state estimation formulation. The numerical stability is guaranteed by using Multifrontal QR factorization.

#### WLS State Estimator

The state estimation problem for a power system with *m* measurements and *n* state variables is usually formulated using the following measurement model [9]:(1)z=h(x)+e,
where *z* is the measurement vector (m×1), *x* is the state variables vector (n×1), h(.) is the nonlinear state estimation function (m×1) that relates the measurements to the state variables, and *e* is the vector of measurement errors (m×1), usually considered as independent random Gaussian variables with zero mean and diagonal covariance matrix $R_{z}$ ($R_{z}=diag$ {$\sigma_{z1}^{2},\sigma_{z2}^{2},...,\sigma_{zm}^{2}$}, where $\sigma_{zi}$ is the standard deviation of measurement $z_{i}$).

Through the classical WLS SE, the state estimate vector $\hat{x}$ is obtained by minimizing the index J(x) given by [9]:(2)$$J\left(x\right)={\left[z-h\left(x\right)\right]}^{T}W\left[z-h\left(x\right)\right],$$
where $W=R_{z}^{-1}$ (the weighting matrix). An iterative process is used to obtain a correction ▵x of the state vector. At each iteration the following linear system is solved: (3)$$▵x={\left[G\left(x\right)\right]}^{-1}H{\left(x\right)}^{T}W▵z\left(x\right).$$ Equation (Equation 3) is known as the Normal equation and the coefficient matrix G(x) is called the Gain Matrix ($G\left(x\right)=H{\left(x\right)}^{T}WH\left(x\right)$).

### 2.3. Decentralized DSSEs

The objective of this paper is to provide conclusive evidence for the advantageousness and feasibility of decentralized monitoring of PDSs, focusing on the development and implementation, on a bench, of one of the decentralized architectures proposed in [7]. This architecture allows for the use of decentralized DSSEs, for instance, the multiarea SE approaches that have already been proposed in the literature. Therefore, this paper does not propose any decentralized DSSE.

Multiarea SEs seek computational performance gains (processing time, memory allocation, processing capacity), by exploring the fact that measurements are obtained from a wide area spread across the electrical network. Such estimation process consists of the separation of the electrical network into sub-areas, in which a local state estimation is formulated for the internal nodes of such sub-areas and with a special treatment for frontier regions (boundaries of each area). Different architectures of multiarea SEs perform the state estimation process separately for each area, and the results refine the estimation of the frontier nodes.

Although the first multiarea state estimation approaches for PDSs presented a large number of iterations to converge and had high coordination latency [18,19], more recent approaches show significant gains in computational performance [20,21,22].

### 2.4. Smart Grids

A smart power grid involves information-based applications made available by increasing automation and communication infrastructure. Among the components present in future power grids are SMs, power devices, energy storage, intelligent appliances, and software infrastructures such as interoperability standards and cybersecurity protocols. The motivations for the development of these applications are network resiliency, performance, and efficiencies generated by automating PDSs [23]. SGs aim to modify the power grid to create a powerful system. They do this by incorporating various control modes and automated functions. SMs are used to record energy consumption and other information such as energy quality indicators and the measurements are sent via upstream messages in response to requests from DOCs. On the other hand, it is possible to receive price signals or even have some level of control that can be incorporated into the SMs. In addition, they can also establish a path of communication with the nearest meters [24].

## 3. Networks and Protocols of Communication

The wireless industry has experienced rapid growth in recent years, with the development of several new applications and technologies, such as personal communication services, satellite communications, broadcasting, high-definition TV, wireless networks, Bluetooth, and others. Some of the most recent leading technologies that are being used in wireless applications, are presented below.

### 3.1. Radio Frequency Technologies

#### Long Range

LoRa is mainly being used to deploy applications in machine-to-machine (M2M) communication and IoT. This is because such deployments can benefit from the low-power specification for communication between electronic devices. In addition, LoRa has an adaptive data rate algorithm that helps to maximize the battery life and the network’s ability to transmit data [25]. LoRa is a Low Power Wide Area Network (LPWAN) communication system that has long-distance transmission capabilities and receives support from companies such as IBM, Semtech, and Actility, which are members of the LoRa Alliance, for the development of the communication network. The objective of LoRa development is to create a low-power network for transmitting data from end devices that transmitted a certain amount of data packets at a given time. The LoRa network, developed by Semtech, is designed to work at a frequency of 433, 868, and 915 MHz (depending on locality) and with a transmission rate between 0.25–50kbps [26].

The typical LoRa-based communication network, as defined in the standard LoRaWAN by the LoRa Alliance, is based on a star topology in which gateways relay messages between end devices and the main network [27]. Overall, LoRa has been approached as a network that will bring benefits to the IoT ecosystem. Thus, LoRa is necessary for communications that are carried out over long distances. When compared to short-range protocols, such as Wi-Fi and Bluetooth, it has been shown to be quite advantageous, although there are some disadvantages in transmission speed and limitations of the size of the payload [25].

LoRaWAN is a new wireless protocol from the IoT family that has recently evolved and is gaining popularity in embedded systems powered by a low-power battery. LoRaWAN operates by transferring small amounts of data at short intervals and long-range [28]. There are many communication technology options for implementing SMs, such as GSM/GPRS, Wi-Fi, Bluetooth, ZigBee, Radio Mesh, and LoRa. Technology GSM/GPRS access is used for three-phase meters because they are used to transfer lots of data such as load profile, final billing, possible breaches, and events that occur in the meters. LoRaWAN is the best choice to be implemented in smart grids, for having long-range and low power, being open-source, and lower cost than GSM/GPRS. The frequency range reserved for the use of communication by LoRaWAN ranges from 902 to 928 MHz in America or 433/868 depending on locality [29].

### 3.2. Cellular Technologies

#### 3.2.1. Long Term Evolution

Long Term Evolution (LTE) is a mobile communication architecture that has previously been designed for wireless device communication and operation on licensed bands. It has high spectral proficiency, excellent data rates, low latency in the exchange of messages, and flexibility in the use of the frequency band. It is a pattern that has been designed by the 3rd Generation Partnership Project (3Gpp). The physical layer of an LTE system has downlink and uplink capabilities and requires high peak transmission rates. The access method that LTE uses is based on Orthogonal Frequency Division Multiple Access (OFDMA) with a mixture of higher bandwidth modulation [30]. Due to its wide coverage, application support, and quality assurance of Service (Quality of Services - QoS), LTE networks are considered a choice suitable for M2M applications that use wireless communication. LTE networks are the most suitable cellular network that can be used for wireless applications due to longevity, low latency and spectral efficiency [31].

The LTE system is based on multi-carrier modulations—FDMA single carrier for uplink and OFDMA for downlink—which provides robustness, flexibility, and adaptability management of radio resources. For this reason, it is more suitable to withstand high data rates than conventional single-carrier solutions or CDMA solutions. In addition, multiple access management is more efficient than IEEE 802.11a CSMA/CA and b, and allows us to easily differentiate the quality of radio service into various terminals for data exchange [32].

#### 3.2.2. Narrow Band

In 2016, the international consortium 3GPP developed a new standard of communication called Narrow Band IoT (NB-IoT) which operates on a frequency channel of a width of 200 kHz. The NB-IoT is a wireless communication standard from the LPWAN family, also developed for M2M applications, which enables sensors and other electronic devices to communicate with each other [33]. Compared to other LPWAN technologies, NB-IoT provides electronic systems with communications at a low cost. It enables the exchange of data in hard-to-reach locations and under conditions that may be challenging for other networks, such as underground sites or locations outside of communities. It uses enhanced mechanisms of energy saving to improve battery life and to support small data transfer. To facilitate the use of NB-IoT networks, network procedures are designed to simplify access complexity [34].

The NB-IoT architecture uses 6 different layer protocols, where they are offered several access controls and system-related features, which improve security, as well as the use of the packet data convergence protocol layer - Packet Data Convergence Protocol (PDCP) and the Middle Access Layer - MAC). The Radio Link Control layer (RLC) is responsible for mobility and also provides security through LTE networks [35].

### 3.3. Network and Communications Protocols for IoT Applications

We are in the era of connected objects; IoT is gaining importance in almost all fields, such as business, industry, electronics, automotive and more. All of the objects in today’s world are connected in one way or another. We can control the lights and equipment in our homes or offices. In industries and other fields, equipment, such as public lighting [28], can be controlled from a remote location. Currently, several protocols have been developed for use in IoT and Smart Grid applications. Each of them within its peculiarities, meet aspects related to the reduction of energy consumption, speed of exchange of data, processing capacity, among other parameters.

#### 3.3.1. Message Queuing Telemetry Transport

The arrival of IoT has allowed most electronic devices to collect, process, and monitor data in real-time. MQTT protocol is one of the most popular protocols for data sharing because it is lightweight and easy to implement. MQTT is suitable for systems where a lot of real-time data is shared. There are two elements of the message sent via Protocol MQTT, they include topic and message. The data is sent from an editor to those who subscribe to a topic published by [36].

The central controller that distributes the message is called the MQTT broker. The MQTT broker forwards, filters, and prioritizes publishing requests from publishers to subscribers. To communicate via the MQTT protocol, the editor (or data generator, e.g., sensor system) should define the two elements described above (message and topic) for the MQTT broker. The content of the message is the string data that the editor wishes to share with subscribers through the MQTT broker. Meanwhile, the topic is a string used by the broker to filter and decide what each subscriber should receive [36].

#### 3.3.2. Rest HTTP

Due to the large volumes of data that are generated and sent across the new electronic devices that use the concept of IoT, it is necessary to build systems of communication and architectures that support various communications in different network environments. One of the most popular communication protocols in computer systems is REST HTTP (Representational State Transfer—REST and Hypertext Transfer Protocol—HTTP). Its use is especially popular in software development projects due to its good computational performance, tested in traditional computer cloud computing [37]. HTTP REST follows the client-server interaction model, which implies that each request is independent. REST uses different verbs; GET, POST, PUT, etc., for features and representations that can be transmitted in different formats such as HTML, XML, or JSON. Following the architecture defined by REST, the HTTP protocol has become widely used as a communication protocol for web services and also for the creation of REST APIs for computational distribution systems [38].

#### 3.3.3. IEC 61850

The IEC 61850 standard is certainly the main reference for substation automation. It has been designed to establish guidelines for the virtual modeling of substations, standardizing methods of communication between devices, and defining system performance requirements. The IEC 61850 is an open standard for Ethernet communication between substations, ensuring interoperability between the various Intelligent Electronic Devices (IED) used. The IEC 61850 standard is not defined as a communication protocol, but defines a set of protocols to be used in electrical power projects, among them are Simple Value, Generic Object Oriented Substation Event (GOOSE), Time Sync, Manufacturing Message Specification among others [39,40].

The GOOSE protocol has been designed specifically for applications in substations. The goose protocol has horizontal communications since messages remain at the same level. GOOSE messages have very strict time requirements because they are often associated with critical applications and replace control cables. As a result, the GOOSE protocol is mapped directly to the link layer, which provides greater speed in communication. The GOOSE protocol does not have any method of confirming a receipt of the message, prioritizing the speed of communication. Therefore, to achieve the appropriate level of reliability, the protocol uses a specific relay scheme, which consists of successive repetitions of the message over a short period. This mechanism is triggered as a result of an event when there is variation in the state of some variable. The GOOSE communication mechanism, where T0 represents the transmission time during stable conditions (without any variation), usually around 1 second. T1, T2, and T3 represent transmission times right after an event This scheme ensures the reliability of communication because if the first message does not reach its destination for some reason, subsequent transmissions will have a high probability of reaching the destination. In the implementation process, T1 = T2 = 4 ms and T3 = 8 ms; these values are considered to be typical of this protocol [39,41].

There are several other communication protocols that have been used to create the so-called smart homes. The smart home concept is based on the interaction between services and resources. This idea results from the convergence of several areas: entertainment, security, energy management, and healthcare. The smart home paradigm can be the answer to these demands, as long as the residence is equipped with technologies that observe the inhabitants and offer proactive services that can provide comfort, security, energy savings, sustainability, and home care. Some of the most widely used communication protocols for these applications are Power Line Communication (PLC) Bluetooth Low Energy, ZigBee, Meter Bus, Z¬Wave and they have technological updates.

## 4. Embedded Platform for the Decentralized Monitoring of Power Distribution Systems

### 4.1. The Implemented Decentralized Architecture

The decentralized architectures for the monitoring of PDSs proposed in [7] are based on ESEMs that manage information and allow for the real-time monitoring of LV networks near the SMs, contrary to the centralized monitoring strategies currently used in DOCs. Among these decentralized architectures, the obtained theoretical results demonstrated that one with ESEMs in the secondary of all distribution transformers was the best in terms of the data volume and computational processing requirements. This architecture is illustrated in Figure 1.

In this architecture, each ESEM installed in the secondary of a distribution transformer is responsible for the state estimation process of the LV network connected to that transformer. The input data of each ESEM module include the topology and the parameters and real-time measurements of the corresponding LV network. Based on these data, each ESEM executes a WLS SE to estimate the state variables of each LV network, i.e., the estimate of the voltage magnitude and phase angle of all buses of the corresponding LV network (each bus corresponds to a Customer Unit (CTU)). After that, each ESEM sends the DOC the load estimation of each distribution transformer (the estimate of the net real and reactive power injection of each distribution transformer). With this information, an SE can be executed in the DOC to estimate the state variables of all feeders and distribution substations.

### 4.2. Hardware and Software Requirements

To emulate an AMI in the laboratory, the modules and communication technologies developed for projects involving IoT applications were used. Figure 2 illustrates the decentralized architecture that was implemented in the laboratory, highlighting the electronic components used.

To emulate the behavior of an SM a Raspberry PI 3 was used in conjunction with a LoRa Ra-01 module to send the measurements every 0.5 s from each CTU to the corresponding ESEM. The code for the management of the LoRa Ra-01 modules, the embedded board settings, and the sending of measurements to the processing gateway was performed using the Python language.

Figure 3 shows the development scheme proposed in this paper for emulation of an AMI in a laboratory.

A load flow calculation gives the reference values for the state variables ($x^{lf}$) and measurements ($z^{lf}$). To simulate measurement values Monte Carlo simulations were performed by including random noise in the reference load flow values [42,43,44].

The *i*-th measurement value was calculated by adding a random noise with Normal distribution $u_{i}∼N\left(0,\sigma_{i}\right)$ in the corresponding *i*-th measurement reference value, according to (Equation 4).
(4)$$z_{i}=z_{i}^{lf}+u_{i}$$

Different precision levels, for each measurement, generate the noise standard deviation $\sigma_{i}$ according to (Equation 5):(5)$$\sigma_{i}=\left|z_{i}^{lf}\right|pr_{i}/3$$
where $pr_{i}$ is the precision of the *i*-th meter. In this research, it was assumed 5% for SM, 2% for SCADA meters, 1% for virtual measurements (passive buses with no generation or load).

For the development of the processing gateway, the ESP32 LoRa module was used as a receiver of the data (measurements) sent by Ra-01. After being received, these data are processed and converted to be sent to the Central Processing Unit (CPU) via the UART interface (CPU represents the DOC that performs the state estimation process to all feeders and substations of the PDS under analysis). With the data (measurement values derived from the emulation of the SMs) available in the Jetson Nano processor, as well as the network topology and parameters of the corresponding LV network, WLS SE is running in the ESEMs, which are programmed in Python language.

With the result of the state estimation process performed in each ESEM, the connections with the TELIT 4G LTE module are initialized. The module is accessed via UART serial communication and AT commands are sent through this communication channel. This establishes the registration of the module of the carrier’s cellular network and enables an IP address to access the internet network, upon request. At the end of the entire initial configuration, it is possible to make an HTTP request and open communication sockets directly to the chosen server and thus forward the data referring to the state estimation result that can be accessed by the DOCs of the concessionaires, being of high importance for the knowledge of the current operating state of the electrical network.

The Things Board IoT platform was used to receive the result data of the state estimation process performed at Jetson Nano. The data is sent in JSON format using the MQTT protocol and stored on the server to be displayed in the developed dashboard, presenting information about the operating state of the PDS under analysis (see Figure 4).

## 5. Validation of a Decentralized Monitoring of PDSs via ESEMs

To validate the decentralized monitoring of PDSs using IoT concepts and the proposed ESEMs, simulations were performed from the implemented AMI prototype.

Figure 5 shows the network topology of the PDS used to carry out the tests. It is a part of the real PDS presented in [45]. The tested PDS has 114 MV buses operating in 13.8 kV, 5 distribution transformers, and 94 LV buses operating in 220 V.

### 5.1. Metering System

To execute the state estimation process in the PDS presented in Figure 5, the existence of SMs was considered in all the CTUs of the LV networks, and also SCADA voltage magnitude measurement in the substation. A total of 99 virtual measurements were also considered.

### 5.2. Accuracy Analysis

As mentioned in Section 4.2, a load flow calculation gives the reference values for the state variables ($x_{lf}$) and measurements ($z_{lf}$). Vector ($x_{lf}$) is used to analyze the accuracy of the state variables estimated by ESEMs using the Relative Mean Error ($EMR_{30}$) and the Standard Deviation ($\sigma_{EMR}$) as given in Equations (Equation 6) and (Equation 7).
(6)$$\begin{array}{c} EMR_{30}=\frac{1}{n_{p}}\cdot \sum_{K=1}^{n_{p}}\left|\frac{{\hat{x}}_{k}-x_{k}^{lf}}{x_{k}^{lf}}\right|\cdot 100 \end{array}$$
(7)$$\begin{array}{c} \sigma_{EMR}=\sqrt{\frac{1}{n_{p}}\cdot \sum_{K=1}^{n_{p}}{\left(\left|\frac{{\hat{x}}_{k}-x_{k}^{lf}}{x_{k}^{lf}}\right|\cdot 100-EMR_{p}\right)}^{2}} \end{array}$$
where, (*K*) denotes the number of buses of the PDSs, (${\hat{x}}_{k}$) the result of the estimation process of 30 samples, and ($x_{k}$) the result of the previously stored estimation process for reference.

To perform the accuracy analysis considering $EMR_{30}$ and $\sigma_{EMR}$, based on $z_{lf}$ 30 measurement samples that had been generated (from Monte Carlo simulations as presented in Section 4.2).

Table 1 presents the comparison between the $EMR_{30}$ and $\sigma_{EMR}$ calculated by a Centralized Architecture and by the implemented Decentralized Architecture. In the centralized architecture, the WLS state estimation process is executed just once considering all the MV and LV network buses. On the other hand, in the decentralized architecture, 5 ESEMs perform a WLS state estimation process on the corresponding LV networks. After that, a WLS state estimation process is performed only on the MV buses, considering the net active and reactive powers estimated by the ESEMs in the distribution transformers.

The mean processing time, for the 30 runs of the WLS state estimation process using the NVIDIA Jetson Nano as CPU, was stored for each LV network present in the test system and is presented in Table 2. It is noteworthy that the decentralized architecture enables parallel execution of the WLS state estimation in each ESEM. Mean processing time for the centralized approach is also presented in Table 2, where all system buses are processed on a single machine. For the centralized architecture, a computer was used with the following technical specifications: Intel Core i3 processor 1.80 GHz RAM 4 GB.

### 5.3. Future Works

The new energy networks bring new challenges to several SG-related areas. The implementation of these networks requires the presence of IoT technologies developed for low consumption and high data exchange applications; SMs, allowing for real-time measurements and improvements in processes carried out for the operation and control of PDSs. In the context of SM implementation in PDSs, supported by disruptive communication technologies, future lines of research will focus on carrying out a detailed study of the communication capacity in relation to the identified problems, such as data transmission and storage capacity or the cybersecurity challenges that would impede correct SG operation.

Therefore, future studies may include the evaluation of several other factors that add robustness to the state estimation process. Among these factors are the interesting determination of the coverage range of wireless technologies to be used, the development of three-phase ESEM for LV networks, and also considering the presence of CTUs monitored and not monitored by SM. In addition, it will be possible to implement a pilot of this proposal and future applications in real LV networks, taking advantage of the creation of AMI concessions and evaluating the development in real projects.

Finally, computer simulations and implementations in research laboratories can be used to validate the three-phase ESEM. Moreover, real electrical systems at power utilities, that are equipped with SMs, may be used.

## 6. Conclusions

Given the growing technological challenges presented by the new SG and the electricity market, it is necessary to implement improvements in the way electric companies manage the information obtained from electrical grids. The complexity of operating and controlling PDSs has increased as a result of the insertion of new electrical grid architectures that allow GS to operate autonomously; the insertion of new renewable energy sources; storage of energy in batteries and an increase in the use of electric vehicles. Such changes make it necessary to leverage new paradigms for the operation of PDSs.

This paper presented a validation of the proposed ESEM, based on the test system created in [45]. The proposed ESEM is based on: the information made available by the SM installed in CTU, which sends consumption data from CTUS using communication networks designed for IoT applications. The information is delivered to an ESEM installed in the distribution transformers. The ESEM runs the SE process, and after it has been finalized, it transmits the result to a web server making use of a cellular communication network.

Considering the installation of SMs in CTUs, the platform implemented in the laboratory has made it possible to process data in a decentralized manner, using electronic modules equipped with low consumption communication, which would be installed in the distribution transformers. In addition, by processing the active and reactive power measurements of each client in the ESEM, the amount of information to be sent to the webserver reduces. The results obtained from using the PDS test evidence that ESEM has performed well, ensuring the feasibility of the implementation of embedded systems in the distribution transformers and allowing the state estimation process to be carried out through the use of a decentralized data processing architecture, instead of the general centralized approach used in DOCs.

## Figures and Tables

**Figure 1 sensors-22-02104-f001:**
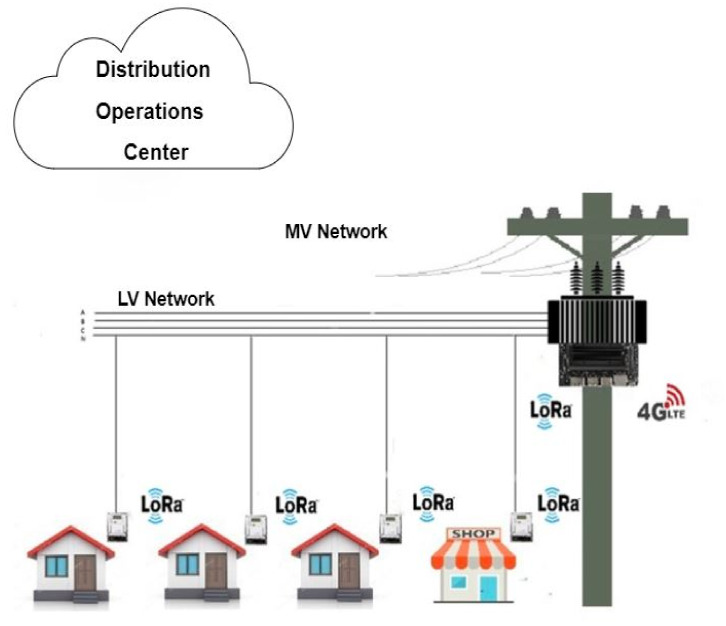
Decentralised Architecture.

**Figure 2 sensors-22-02104-f002:**
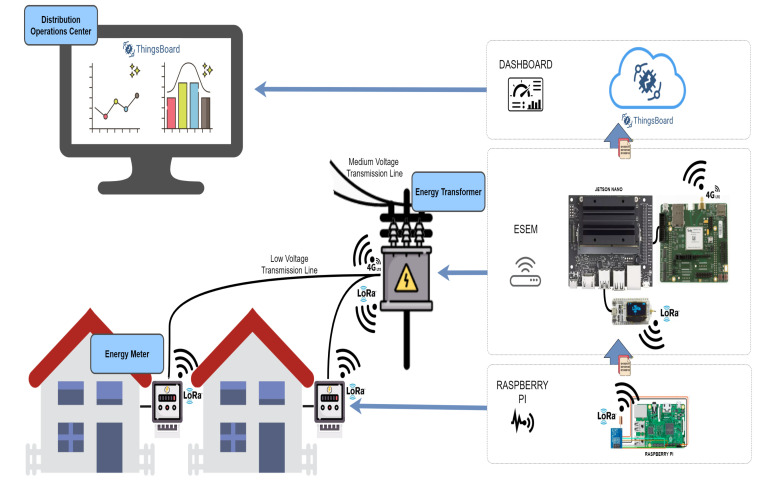
Implementing the chosen Decentralised Architecture.

**Figure 3 sensors-22-02104-f003:**
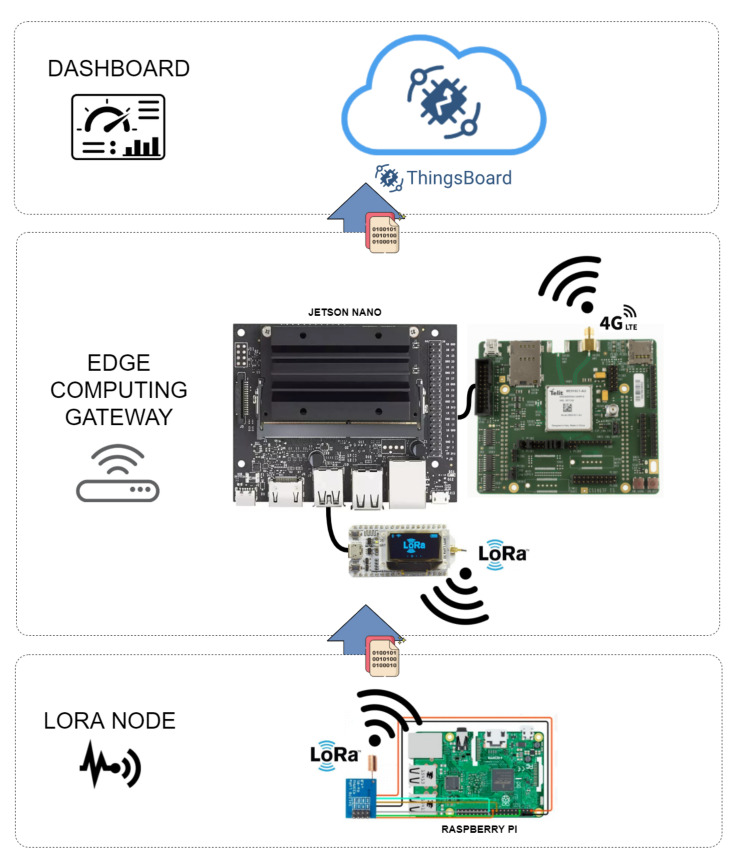
AMI schematic.

**Figure 4 sensors-22-02104-f004:**
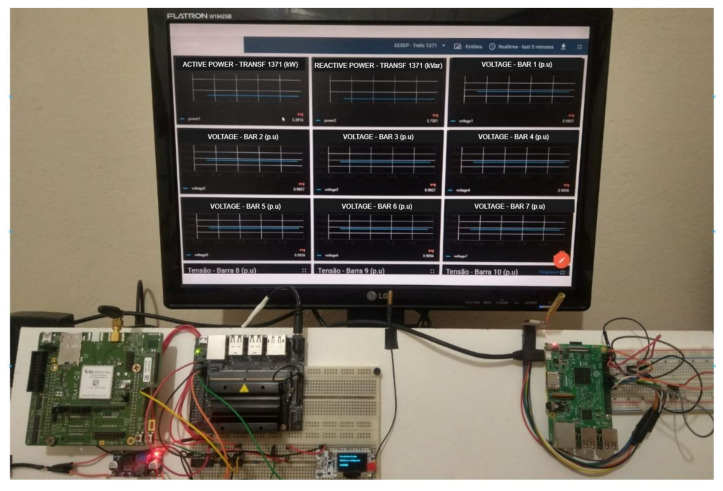
ThingsBoard Dashboard and electronic devices.

**Figure 5 sensors-22-02104-f005:**
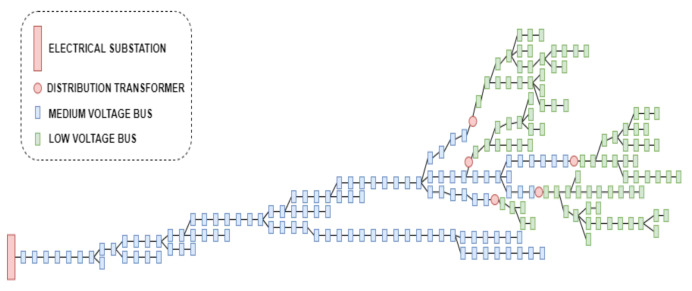
Network Topology of the PDS used to carry out the tests.

**Table 1 sensors-22-02104-t001:** Result of the state estimation process for the complete network.

EMR Centralized	Standard Deviation Centralized	EMR Decentralized	Standard Deviation Decentralized
0.0255	0.0072939	0.0297	0.008119

**Table 2 sensors-22-02104-t002:** Execution time of the state estimation process.

Electricity Network Used	Processing Time (S)	CPU
Complete (MV and LV Buses)	46.4579	Core i3
Primary (MV Buses only)	35.7633	Jetson Nano
LV Buses connected to Transformer 2960	0.3380	Jetson Nano
LV Buses connected to Transformer 1370	1.409	Jetson Nano
LV Buses connected to Transformer 1371	0.7472	Jetson Nano
LV Buses connected to Transformer 1372	1.3451	Jetson Nano
LV Buses connected to Transformer 1373	1.1043	Jetson Nano

## Data Availability

Not applicable.

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
