# Peer review of "Validation of Embedded State Estimator Modules for Decentralized Monitoring of Power Distribution Systems Using IoT Components"

_sensors, 2022, doi:10.3390/s22062104_

Round 1
Reviewer 1 Report
The work is very interesting and relevant, it presents a software and hardware architecture applied to smart grids. In the following paragraphs, I point out a few problems that authors can look at one by one. Among them, I highlight the difficulty I had in understanding the architecture used, in addition to some other errors, such as Table 2, which has only one column and should have two. Because of this error, I will recommend to reconsider after major revision. Also, for validation from runtimes, due to the error in Table 2, I could not see runtimes for the Intel Core i3 CPU. The goal was to compare with the result of Jetson Nano, right? I am so far not convinced of the need for a Jetson Nano GPU for this application. One objective for this project to be able to be moved forward has to be the cost, and the Jetson Nano will make the system very expensive. Couldn't a Raspberry Pi simply calculate the WLS? For the validation, I believe this work should show the runtimes of three architectures: The Jetson Nano, the Intel Core i3 mentioned in the text, and a Raspberry Pi 3 or 4.
Another flaw in this article is in the writing of Section 3. It mentions many technologies that are not used in this work. I think that this theoretical basis should stick only to what is necessary for the understanding of this paper.
There are some similarities in the text with the reference [3] written by the same authors, but it is only on the theoretical basis and acceptable.
As will be discussed in many places in the paragraphs below, it is necessary to better describe the architecture in the Introduction. The solution requires an improvement not only in the text but also in Figures 1, 2, and 3.
Apart from these flaws, the work is quite relevant and deserves to be published. I congratulate all the authors for their excellent work. Below are the technical remarks made by this reviewer, and then annotations of errors in English. Such English corrections are suggestions and are not committed to exhausting all errors in the text. I ask the authors to proofread to find any errors.
Technical remarks:
* The title of the paper is a little big and the only acronym that was not written out in full was AMI, and precisely this one is ambiguous in the context of IoT. There is also the acronym AmI which stands for Ambient Intelligence. So, reading the title, it is not possible to deduce the theme of the paper at first glance. There is also no mention of power distribution, making it more difficult to know the area of ​​the paper just by the title. The word Power is missing from the title. This is just a comment, I don't intend to ask the authors to change the paper's title, maybe that's not even possible, but other ideas would be:
An IoT Based Advanced Measurement Infrastructure Prototype for Decentralized Monitoring of Power Distribution Systems
Validation of an IoT Based Advanced Measurement Infrastructure Prototype for Decentralized Monitoring of Power Distribution Systems
Validation of Embedded State Estimator Modules for
Decentralized Monitoring of Power Distribution Systems Using IoT Components.
Or something very simple like:
IoT Based Decentralized Monitoring of Power Distribution Systems
* In keeping with the original title, the Sensors standard is to have all words capitalized, except prepositions. So, correct the words "based" and "concepts".
* Put the acronym AMI in the abstract, on line 7.
* Describe further in the Introduction where each NVIDIA Jetson Nano and Raspberry PI 3 were placed and how many were used. The reader is concerned that every SM will need one. Indeed, the Introduction needs to give a better sense of the architecture we are dealing with. Put the LoRa also in the description.
* In the paragraph of Line 98, the authors state, using the past tense, that the diffusion of electric vehicles has changed the passive behavior of PDSs. However, electric vehicles are not yet a reality. And even if, say, 80% of vehicles around the world were electric, how much would that weigh on the distribution network? This seems like a controversial issue. Can you cite a reference from any study on the impact of electric vehicles on electricity distribution? In my view, the distribution system grows faster than the diffusion of electric cars. Also, review the verb tense used in the paragraph.
* Do Equations 2 and 3 refer to the conventional WLS approach? What is a reference for this? Perhaps the reference [1] that you cited in [3] (Primadianth) will do.
* In Line 163, you affirm that you are going to present technologies for high data transmission rates. There you present LoRa. The LoRa technology is not for high data rates. It is on the order of bits per second, 0.25 to 50 kbps as quoted later in the text. Therefore, the authors should change the introductory sentence in line 163. The objective of LoRa is to transmit, with low data transmission rates, but over long distances and with low power consumption. And then there's the NB-IoT. Put something like high data transmission rates and/or long transmission distances.
* Section 3 has different subsections with the same goals. For example, 3.1.1 deals with LPWAN, and 3.3.1 deals with LoRaWAN, which are the same thing. So I think Section 3 can be better organized. Furthermore, will everything mentioned in Section 3 be used in this work? Does this work use MQTT?
* In Figure 2, why does the Raspberry Pi need to communicate with ESEM via LoRa? Isn't SM a component that is integrated into ESEM? Can't the Raspberry Pi stay together directly connected to the ESEM? Maybe that's it, but the drawing is a little deceiving. Is the Raspberry Pi LoRa just for communicating with CTUs? The same question is in Figure 3.
* In Figures 2 and 3, write the name Jetson Nano next to the image.
* In my view, it seems that there is an excess of components in this architecture. Does Jetson Nano not communicate with CTUs via LoRa? So why the Raspberry Pi? In my opinion, you don't need such computational power as a GPU in this application.
* Lines 319 to 326.
I don't understand who is the DOC in this architecture? Is it the Jetson Nano? It will be necessary to improve the figures.
* Figure 5 is completely unnecessary as none of these graphs belong to the research. And the illustration of this screen has already appeared in Figure 4.
* Line 349:
"SMs in all the CTUs of the LV networks"
Do you mean a Raspberry Pi on every customer? This becomes very expensive and unnecessary. An ESP32 module with LoRa does the job.
* Line 377:
Mean processing time for the centralized approach is also presented in Table 2.
At where? There is only one column in this table. By the way, please put in the legend if these runtimes are CPU or Jetson runtimes
Line 380:
Intel Core i3 180GHz processor and 180GHz RAM 4GB
180GHz? Oh my God!
Please review this specification.
Notes on English
* real-time has the hyphen. In the paper, some are with and others without. Please correct. Search the entire article. On lines 279, 288 the error also appears.
* Line 91:
customer´s energy demands
=>
customer´s energy demand
* The following sentence has problems and is difficult to read:
To produce the ESEM electronic development equipped with high-performance processors with Linux operating system is used to enable matrix analysis related to the Weighted Least Squares (WLS) state estimation process.
To have "parallel construction" with the previous sentence, which uses "Development and implementation", change it to:
Development (or Production) of the ESEM electronic equipped with high-performance processors with Linux operating system to enable matrix analysis related to the Weighted Least Squares (WLS) state estimation process.
* When you write "mx1" do not use the letter "x" but instead use the Latex command \times in math-mode: $m\times 1$, otherwise it confuses with state variable x.
* Line 154:
and the measurements are send upstream messages
=>
and the measurements are sent via upstream messages
* Line 155:
In another hand, there is the possibility for receive price signals or even some level of control can be incorporated into the SMs.
=>
On the other hand, there is the possibility of receiving price signals or even some level of control that can be incorporated into the SMs.
* Line 168:
In addition, LoRa has a adaptive data rate
=>
In addition, LoRa has an adaptive data rate
* Line 169:
algorithm helps to maximize the battery life
=>
algorithm that helps to maximize the battery life
* Line 188:
to operate on the bands Licensed.
=>
to operate on the licensed bands.
* Line 191:
The physical layer of a system LTE has
=>
The physical layer of an LTE system has
* Line 196:
QOS => QoS
* Line 197:
LTE networks are the most suitable cellular networks
=>
LTE networks are the most suitable cellular network
* Line 227: "fine connected" is a bit of an exaggeration:
objects in today’s world are fine connected to each other in one way or another.
=>
objects in today’s world are connected in one way or another.
* Line 282: Be more detailed: secondary winding
in the secondary of all distribution transformers
=>
in the secondary winding of all distribution transformers
* Line 356:
equations => Equations
* Line 374
presents => present
* Line 377
time.Mean => time. Mean
* Line 377:
i.e. in the same period of time.
It is not necessary to explain the meaning of parallel execution
* Line 398: Rewrite this phrase:
and can evaluate the development in real projects
Author Response
Response to the Editor and Reviewers:
Manuscript ID: sensors-1527093
Rosvando Marques Gonzaga Junior, Sergio Márquez-Sánchez, Jorge Herrera Santos, Rodrigo Maximiano Antunes de Almeida, João Bosco Augusto London Junior and Juan Manuel Corchado Rodríguez.
We thank the editor and the reviewers for their insightful comments and requests regarding the paper. We carefully analyzed all the comments, which allowed us to make improvements and refinements in a new version. A response point-by-point to the reviewers' comments is presented in the following in blue. All manuscript updates are also highlighted in blue in this revised version of the manuscript.
We trust the changes introduced in the manuscript can meet the requirements of the Journal.

Reviewer 2 Report
First of all, I suggest reducing the title.
In the abstract, it is not clear the contribution of this work in front of state of the art.
The real data validation through simulation is relevant and interesting for the community.
Lora is a long-range technology. The authors need to discuss about the other technologies without long range for houses and other environments ner to the data collection.
Many pages, e.g., 10 and 11, has wrong format with only one figure or a short fragment of text.
Fig. 6 is difficult to see.
The authors need to discuss about the GOOSE messages and the IEC 61850 requirements (e.g., temporal requirement).
Author Response

(The authors gave the same response as above.)

Reviewer 3 Report
The paper proposes: (i) Development and implementation, in laboratory, of an Advanced Measurement Infrastructure prototype to enable the simulation of a smart grid. To emulate the information traffic between smart meters and distribution operation centers, communication modules that enable the use of wireless networks for sending messages in real-time are used, bringing concepts from both IoT and Edge Computing; and (ii) Development and implementation, in laboratory, of a decentralized architecture based on Embedded State Estimator Modules (ESEMs) that manage information from smart meters at lower voltage networks to perform real time state estimation in PDSs. Simulations performed on a real PDS with 208 buses (considering both medium and low voltage buses) have validated the paper purposes.
The topic fits the scope of the journal.
The manuscript is well written, the structure of the paper is clear and the language is proper.
The introduction section must be revised, rewrited and reorganised in order to clarify the motivation, objectives and the contributions in which advances the related work. It is not really clear the contributions compared to state of the art.
I strongly suggest revisiting the up to date references in the topic covered by the paper.
Figures need to be improved to better quality.
The manuscript needs a revision in order to correct many typos.
Author Response

(The authors gave the same response as above.)

Round 2
Reviewer 1 Report
Table 2 has been corrected and it is now possible to understand its purpose. The Introduction and the figures were revised, resolving all initial doubts in the first version. Therefore I am satisfied with the text of the second version of the paper.